# The Childhood Cancer Diagnosis (CCD) Study: a UK observational study to describe referral pathways and quantify diagnostic intervals in children and young people with cancer

Dhurgshaarna Shanmugavadivel [iD],[1] Jo-Fen Liu [iD],[1] Ashley Ball-Gamble [iD],[2] Angela Polanco [iD],[2] Kavita Vedhara [iD],[1] David Walker,[3] Shalini Ojha [iD] [1,4]

[1]Academic Unit of Population and Lifespan Sciences, University of Nottingham Faculty of Medicine and Health Sciences, Nottingham, UK
[2]Chief Executive Office, Children's Cancer and Leukaemia Group, Leicester, UK
[3]Children's Brain Tumour Research Centre, University of Nottingham, Nottingham, UK
[4]Children's Hospital, University Hospitals of Derby and Burton NHS Trust, Derby, UK

**Correspondence to**
Dr Dhurgshaarna Shanmugavadivel;
shaarnashan@doctors.org.uk

## ABSTRACT

**Introduction** Childhood cancer is diagnosed in 400 000 children and young people (CYP) aged 0–19 years worldwide annually. In the UK, a child's cumulative cancer risk increases from 1 in 4690 from birth to aged 1, to 1 in 470 by age 15. Once diagnosed, access to treatments offers survival to adulthood for over 80%. Tumour diagnoses are at a later stage and mortality is higher when compared with those in other parts of Europe. This means higher risk, more intensive therapies for a cure. Some CYPs are known to experience delays to diagnosis which may further contribute to poor outcomes. This study aims to understand the current pathway of childhood cancer referrals and diagnosis and quantify diagnostic intervals in the UK.

**Methods and analysis** This is a prospective multicentre observational study including all tertiary childhood cancer treatment centres in the UK. CYP (0–18 years) with a new diagnosis of cancer over the study period will be invited to participate. Data will be collected at initial diagnosis and 5 years after diagnosis. Data will include demographic details, clinical symptoms, tumour location, stage and clinical risk group. In addition, key diagnostic dates and referral routes will be collected to calculate the diagnostic intervals. At 5 years' follow-up, data will be collected on refractory disease, relapse and 1-year and 5-year survival. Population characteristics will be presented with descriptive analyses with further analyses stratified by age, geographical region and cancer type. Associations between diagnostic intervals/delay and risk factors will be explored using multiple regression and logistic regression.

**Ethics** The study has favourable opinion from the York and Humber, Leeds West REC (19/YH/0416).

**Dissemination** Results will be presented at academic conferences, published in peer-reviewed journals and disseminated through public messaging in collaboration with our charity partners through a national awareness campaign (ChildCancerSmart).

**Study registration** researchregistry.com (researchregistry5313).

## Strengths and limitations of this study

► The first nationwide study using prospective point of care data to map childhood cancer diagnostic pathways with measurements of diagnostic intervals.
► It includes the whole spectrum of cancers in children and young people aged 0–18.
► It will collect social, demographic and clinical data prospectively to reduce recall bias and explore associations of diagnostic intervals with these characteristics.
► Diagnostic interval will be calculated from the point of symptom onset. However, these data will necessarily be retrospective and may, therefore, be affected by recall bias.

## INTRODUCTION

Childhood cancer is diagnosed in 400 000 children and young people (CYP) 0–19 years worldwide annually.[1] Contrary to popular belief, childhood cancer is not rare.[2] In the UK, the individual risk of cancer from birth to age 15 years is 1 in 470[2] with 1645 new cases in 0–14 year olds and 2110 new cases in 15–24 year olds diagnosed each year.[3] Importantly, the incidence of childhood cancer has increased by 15% since the 1990s with a slightly higher incidence in boys than girls (in under 15s: boys, 1 in 420; girls, 1 in 490).[3] While genetic predispositions are well documented, no modifiable or preventable risk factors have been identified.[4]

Childhood cancer is also the largest illness cause of death in CYP globally, and in the UK, responsible for over 1 in 5 deaths among 0–15 year olds.[3] As such, in 2018, the WHO identified childhood cancer as a global disease burden and launched the Global Initiative for Childhood Cancer aiming

to improve survival rates to 60% by 2030, saving over 1 million lives.[5]

The overall 5-year survival estimate in the UK is 84% across all childhood cancers, a statistically significant increase from 77% in 2001.[3] The improving cure rates over past decades have been achieved by the introduction of expertly delivered, complex therapies. Despite this, the UK performance for stage distribution at diagnosis for multiple tumours and outcomes compares unfavourably to those in leading European countries and survival rates are worse than in other countries, for example, Iceland has a 90.1% 5-year survival rate.[6–8] A possible cause for the poorer outcomes is delay in diagnosis, the reasons for which may be multifactorial. Previous studies have reported diagnostic pathways for CYP with cancer in England using data reported to central registries.[9] Children (0–14) were found to present more commonly via an emergency presentation than those aged 15–25 or 26–44, however, this did not seem to cause a significant disadvantage in survival outcome.[9]

Symptoms in children are often non-specific, mimicking more common ailments. Furthermore, the perceived rarity of childhood cancer means it is often not considered as a diagnosis until there are multiple symptoms by which time the disease is at a more advanced stage. Despite a systematic review confirming that CYP experience delays to diagnosis,[10] there is a dearth of research exploring how and why such delays occur.

In the absence of recognised modifiable risk factors or feasible screening strategies, the most effective approach to improving patient outcomes is early diagnosis that may enable prompt, effective treatment. Childhood cancer survivors are left with long-term effects, or late effects, caused by either the cancer itself or its treatment.[11] Late effects include problems with growth, organ function, fertility, cognition and academic achievement.[12] It has been reported that two-thirds of childhood cancer survivors will develop at least one late-onset therapy-related complication.[13] Delays in diagnosis add further avoidable disabilities and increased risk of local tumours needing more extensive surgery, for example, amputation versus bone preserving surgery, partial nephrectomy versus total nephrectomy or liver resection versus liver transplant. Furthermore, advanced disease requires more extensive radiation fields with greater volumes of tissue irradiation with attendant risk for impaired tissue growth, focal brain/endocrine tissue damage and enhanced second tumour risk. Early diagnosis can therefore reduce mortality and morbidity from the cancer itself and from the intensive burden of the curative treatment required to treat more advanced stage disease.

While it is recognised globally that early diagnosis is crucial and that delays in diagnosis occur, we need to understand the current diagnostic pathway patterns and identify areas of potential improvement to enable improved care.

**Figure 1** A map of all Principal Treatment Centres in the UK courtesy of Children's Cancer and Leukaemia Group (CCLG).

## AIMS AND OBJECTIVES

The aim of this study is to understand the diagnostic intervals (DIs) and referral pathways for CYP diagnosed with childhood cancer in the UK. The study objectives are: In CYPs with a new diagnosis of childhood cancers
1. To determine the DIs.
2. To determine the route of referral.
3. To analyse the differences in DIs and routes of referral between cancer types, age of presentation and geographical region.
4. To explore the associations between DIs and patient and disease characteristics.

## METHODS AND ANALYSIS
### Study design and setting
This is a prospective multicentre observational study including all tertiary childhood cancer treatment centres, that is, Principal Treatment Centres (PTCs) for Paediatric Oncology and Haematology in the UK (figure 1).

### Participant eligibility
All CYPs aged 0–18 years with a new diagnosis of childhood cancer over the study period will be invited to participate (table 1). This age group was chosen to correlate with the CYP cared for by paediatric clinical services within the UK.

### Study procedures
CYP will be recruited from all PTCs across the UK. Recruitment will be supported by the Children's Cancer and Leukaemia Group (CCLG) who have an established research network, with a Principal Investigator responsible for the study at each site (online supplemental file

**Table 1** Criteria for participant inclusion and exclusion

| | |
|---|---|
| Inclusion criteria | Children and young people at age 0–18 years<br>**AND**<br>A new diagnosis of a childhood cancer (see online supplemental file 1 for complete list)<br>**WITH**<br>► the ability for their parent/guardian to give informed consent if age of the child is less than 16 years of age<br>► **Or** the ability for the young person to give informed consent if 16–18 years of age<br>► **Or** a consultee/legal representative is available to provide an opinion/consent if the young person is aged 16–18 and is deemed to lack capacity to consent for themselves. |
| Exclusion criteria | Age at diagnosis over 18 years of age<br>Patient diagnosed with cancer outside the UK |

3). The study opened to recruitment on 30 September 2020 and is on the National Institute of Health Research Portfolio.

## Recruitment

Eligible participants will be recruited following a confirmed diagnosis of any cancer at a PTC. In the UK, once the CYP is referred to the PTC with a diagnosis of cancer, they have a consultation with their oncology care team. During this first consultation, a full history of the events leading up to the diagnosis is recorded. A member of the clinical care team will identify eligible participants at this consultation. Recruitment will occur via two possible methods (A or B) (figure 2) to maximise participation, provide flexibility to recruiters and potential participants and give CYP the optimal opportunity to participate.

## Informed consent

All participants will provide written informed consent.

### Method A: in-person consent via paper forms

Participants who are inpatients on hospital wards will be given the study information and opportunity to discuss participation with a researcher. A researcher will seek

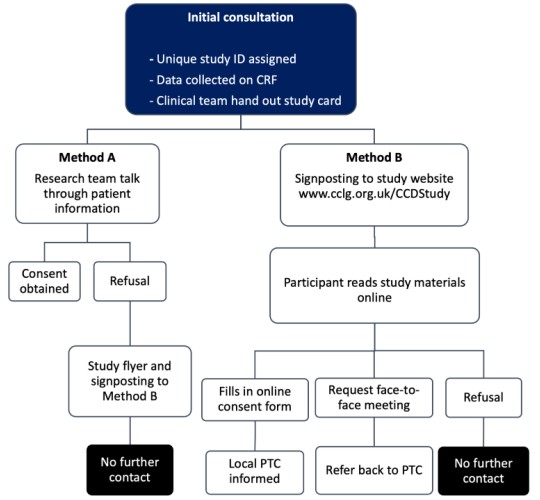

**Figure 2** Recruitment methodology for the study. CCD, Childhood Cancer Diagnosis; CRF, case report form; PTC, Principal Treatment Centre.

written informed consent after at least 24 hours of the participant having received the study information.

### Method B: consent using online forms

During the initial consultation, a study flyer including the study title, a brief explanation of the study and contact details of the study team will be offered to potential participants. The participants can access the study website, read the information, discuss with the research team if they wish and give written informed consent via the study website (www.cclg.org.uk/CCDStudy).

Both pathways will be followed in keeping with the principles of Good Clinical Practice.[14] Participation will be entirely voluntary, and treatment and care will not be affected by the decision. It will also be explained the participant can withdraw at any time, but attempts will be made to avoid this.

For <16-year-old CYP, consent will be obtained from the parent/guardian. Those between 16 and 18 years of age can provide consent. Involvement of the parents in decision-making will be encouraged unless the young person objects to this involvement. For those aged 16–18 years old who lack capacity to consent, a consultee or legal representative will be consulted and asked to consent in keeping with the Mental Capacity Act (England and Wales); the Adults with Incapacity (Scotland) Act or the Mental Capacity Act (Northern Ireland) 2016.

## Data collection

Data will be collected by the clinical care team after recruitment from the first consultation at the PTC when the initial cancer diagnosis is made. Further follow-up data will be collected 5 years after the initial diagnosis. All PTCs that treat CYP with cancer across the UK, through our collaboration with the CCLG, will participate. Data will be collected on standardised case report forms (online supplemental file 2).

Data will include demographic details and characteristics such as sex, age, ethnicity and Index of Multiple Deprivation (IMD)[15] (calculated from home postcode without any health domain component). Clinical signs and symptoms at diagnosis, tumour location, tumour stage and clinical risk group (if applicable) will be collected. The International Classification of Childhood Cancer (ICCC-3)[1] will be used to code the diagnoses and the Toronto Paediatric Cancer Stage Guideline will be used to record tumour stage. These classification systems were chosen as they are internationally accepted and will therefore allow comparison with other studies.

## Primary outcome measure

The primary outcome measure is the total diagnostic interval (TDI), as defined in the literature (table 2).

## Secondary outcome measures

The secondary outcome measures are the patient interval (PI) and the DI (table 2).

**Table 2** Definitions for diagnostic intervals[16]

| Diagnostic interval | Definition |
| --- | --- |
| Total diagnostic interval (TDI) | Time from symptom onset to the time diagnosis was established (sum of PI and DI) |
| Patient interval (PI) | Time from symptom onset to the time of first consultation with a healthcare professional |
| Diagnostic interval (DI) | Time from first consultation with a healthcare professional to the time diagnosis was established |

## Data for calculating DIs

To calculate the DIs, three key dates will be collected: date of symptom onset, date of first presentation to healthcare and date of diagnosis (clinical, imaging, biopsy). The date of symptom onset will be determined by the participant or their parent/guardian. Thus, it will necessarily be retrospective and self-reported. The date of first presentation to healthcare will be defined as the first presentation to any healthcare service with signs/symptoms attributable to the tumour as reported by the participant or their parent/guardian. The date of diagnosis is defined as the day when the cancer diagnosis was established, clinically, radiologically or histologically as recorded in the participants' medical records at the PTC. It will include the dates of clinical diagnosis, imaging, biopsy, histopathology report and/or multidisciplinary team meeting where the diagnosis was established.

Where exact dates, such as participant reported dates, cannot be established, approximates will be used. If the date is specified to the nearest week, it will be assumed to be the Monday at the start of the week. If specified to the nearest month, it will be recorded as the first day of the month and if specified to the nearest season, it will be recorded as the first day of April for 'spring', July for 'summer' or 'mid-year', October for 'fall' or 'autumn'. In winter, attempt to determine whether the diagnosis was 'late in the year' (use December with the applicable year) or 'early in year' (use January with the respective year). Missing dates will be recorded as 1 January 1900.

## Route to referral

To map out the route of referral, five key pieces of information will be collected: the first healthcare professional that the participant consulted about relevant symptom(s); the number of healthcare visits between onset of symptoms and diagnosis; the patient's place of care when the investigation that identified the tumour was requested; whether the diagnosis is an incidental finding and the source of referral leading to diagnosis.

## Planned 5-year follow-up

Centres will be approached at 5 years after the close to recruitment and asked to submit dates of first relapse, refractory illness (by date of pathology or imaging) and/ or death to calculate 1-year and 5-year survival. A separate follow-up protocol will be written in due course.

## Sample size and justification

This is an observational study of all incident cases of childhood cancer over a 2-year period. There are 1645 new diagnoses of cancer in the under 15 age group each year in the UK, and 2110 new diagnoses in the 15–25 age group each year in the UK.[3] As we are studying the 0–18 age group, we anticipate approximately 2000 new cases per year. Over 2 years, this would be 4000 new cases. Based on our previous experience, with a 70% recruitment rate, we expect around 2800 cases in the study period.

## Statistical analysis

Descriptive analysis will be used to characterise the study population. Data will be presented as mean and SD or median and IQR for continuous data and as counts and percentages for categorical data.

## Diagnostic intervals

The three DIs (TDI, PI and DI) will be calculated and reported as median (IQR) as defined in the literature.[16]

Subgroup analyses stratified by age, sex, geographical region, socioeconomic status and cancer type will be performed. Student's t-test, $\chi^2$ or Kruskal-Wallis tests will be used for comparison between groups as appropriate.

Clinical factors of interest are tumour type, tumour location and presentation symptom. Outcome variable of interest is DIs (as continous) and diagnostic delay (diagnostics intervals categorised using percentile cut points). Multiple regression and multivariable logistic regression will be used to estimate adjusted regression coefficients and adjusted ORs for each clinical factor, respectively. Univariable and full clinical model will be fitted, and relationships of all variables including will also be assessed in order to select the variables to be included in the final parsimonious adjusted model. Effect modification with sociodemographic factors (age, sex, geographical regions, socioeconomic status as represented by IMD calculated by resident postcode as a categorical variable) will be explored as appropriate.

All statistical analyses will be conducted using statistical software Stata 16 SE (StataCorp LLC) and/or R studio (RStudio, PBC, Boston, MA). A p value of<0.05 will be considered statistically significant in all analyses.

## Missing data

Where two or more key dates are missing despite these measures, the participant will not be included in the DI analysis. The number of such participants will be reported in the study flowchart and summary statistics comparing participants who had missing data with those whose full data set were available will also be reported. We will not be doing any multiple imputation. For each analysis, numbers missing will be reported.

## Patient and public involvement

The study was designed in collaboration with our charity partner, the CCLG and our parent advisor (AP) who played a key role in shaping the proposal. AP was involved throughout the study design process, including the consent process, and reviewing the patient information leaflet materials. In addition to this, members of the Paediatric Oncology Reference Team (PORT) who are an independent body of parents with experience of childhood cancer, also advised on the study protocol and revised patient-facing documents. Members of PORT sit on the National Cancer Research Institute's Children's Cancer and Leukaemia Study group and regularly advise on research studies.

## Ethical approvals

The study was given a favourable opinion by York and Humber, Leeds West REC (19/YH/0416) on 27/02/2020 and will be conducted in accordance with the ethical principles that have their origin in the Declaration of Helsinki, 1996; the principles of Good Clinical Practice and the UK Department of Health Policy Framework for Health and Social Care, 2017.[14]

## Protocol registration

This study has been registered on researchregistry.com (researchregistry5313).

## DISCUSSION

This is the first national observational study to measure DIs and referral pathways for CYP. The data obtained will allow us to understand the current picture of childhood cancer diagnosis across the UK and identify factors associated with diagnostic delays. It will highlight areas with need for improvement where targeted public health interventions or larger policy changes could be implemented to enable earlier diagnosis. The WHO global effort to improve survival rates by 2030 has led to an urgency to understand the current picture and drive change to meet this ambitious but achievable target.

This national observational study follows from the success of the UK HeadSmart, early diagnosis of brain tumours campaign.[17] The HeadSmart public and professional awareness campaign was launched in 2011 in the UK, aiming to raise awareness of the signs and symptoms of brain tumours in children due to the long DIs. The campaign has been associated with a reduction in the TDI from 14.4 weeks in in 2006 to 6.5 weeks in 2015.[17 18]

This experience, where the impact of the public and professional awareness campaign (www.headsmart.org.uk) was shown to accelerate brain tumour diagnosis, justifies this project. It will generate evidence to better understand the current pathway of childhood cancer referrals and diagnosis and quantify DIs in the UK. The study will primarily inform UK practice but may be used as a model worldwide, as part of WHO global challenge to level up outcomes for children with cancer.

## Strengths

This study will recruit from all PTCs in the UK. This network of PTCs will allow maximal national coverage and give every CYP with a new diagnosis of cancer to participate. As childhood cancer is not treated by other services in the UK, this study will represent the whole UK population.

The collected data will allow analyses of DIs and referral routes as well as their associations with social and clinical characteristics such as age, tumour type, geographical location and tumour stage at presentation. Furthermore, the 5-year follow-up will enable analyses of associations between DIs and refractory disease, relapse and 1-year and 5-year survival providing insight into whether delays in diagnosis affect survival.

## Limitations

The DIs will be collected from dates obtained from CYP and their families. There is a possibility of recall bias. We aim to minimise this by ensuring that the data are collected on the CYP's first presentation at the PTC where a thorough clinical history is recorded routinely. Adequate training will be provided to each site and all reporting clinicians will be given advice on how to record these dates as accurately as possible. All other data including data on outcome will be collected prospectively thus maximising accuracy.

## Dissemination of results

This study is a collaboration between the University of Nottingham and the CCLG. The results will be disseminated to healthcare professionals through conference presentations and peer-reviewed journal publications. In addition, the results of the study will inform health policy makers in the UK to design and implement referral pathways that are improved and informed by this evidence. The data will also be disseminated through public messaging, raising awareness of the signs and symptoms of childhood cancer through a national awareness campaign called Child Cancer Smart.

**Acknowledgements** We would like to acknowledge all site principal investigators at the PTCs. We are particularly grateful as many sites were set up by research teams who continued to recruit and collect data in the midst of the COVID-19 pandemic.

**Contributors** DS, SO, J-FL and DW conceived the study. DS, J-FL, DW, AP, SO and KV were involved with study design. DS drafted the manuscript which was revised by SO and reviewed and edited by all authors.

**Funding** This work is supported by the National Institute of Health Research (NIHR), through a Doctoral Research Fellowship for DS, grant number (DRF-2018–11-ST2-055). It is also supported by The Children's Cancer and Leukaemia Group, award number (CCLG CCS 001/CCLG CCS 002).

**Map disclaimer** The inclusion of any map (including the depiction of any boundaries therein), or of any geographic or locational reference, does not imply the expression of any opinion whatsoever on the part of BMJ concerning the legal status of any country, territory, jurisdiction or area or of its authorities. Any such expression remains solely that of the relevant source and is not endorsed by BMJ. Maps are provided without any warranty of any kind, either express or implied.

**Competing interests** None declared.

**Patient consent for publication** Not applicable.

**Provenance and peer review** Not commissioned; externally peer reviewed.

**ORCID iDs**
Dhurgshaarna Shanmugavadivel http://orcid.org/0000-0002-1912-4543
Jo-Fen Liu http://orcid.org/0000-0001-5796-7878
Ashley Ball-Gamble http://orcid.org/0000-0002-0708-0918
Angela Polanco http://orcid.org/0000-0002-4619-0773
Kavita Vedhara http://orcid.org/0000-0002-9940-7534
Shalini Ojha http://orcid.org/0000-0001-5668-4227

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
