## [Reviewer comments · BMJ Open]

ARTICLE DETAILS

TITLE (PROVISIONAL)	The Childhood Cancer Diagnosis (CCD) Study: a UK observational study to describe referral pathways and quantify diagnostic intervals in children and young people with cancer.
AUTHORS	Shanmugavadivel, Dhurgshaarna; Liu, Jo-Fen; Ball-Gamble, Ashley; Polanco, Angela; Vedhara, Kavita; Walker, David; Ojha, Shalini

VERSION 1 – REVIEW

REVIEWER	Nepogodiev, Dmitri University of Birmingham
REVIEW RETURNED	02-Nov-2021

GENERAL COMMENTS	This protocol is for a study that addresses an important topic and I anticipate that it will produce results that can inform both policy and future research priorities. There are some areas that could be improved - main points: 1. No primary outcome is defined. Three 'diagnostic interval' outcomes are described but it is unclear which is the main primary outcome and which are the secondary outcomes - this should be clarified.2. Statical analysis.-The sentence "This is an observational study and therefore there will be no hypothesis testing" should be deleted as you are planning hypothesis testing (also this statement is non-sequitur).- "In the absence of a standard definition of diagnostic delay , median and 75th percentile will be used as cut-offs". This sounds very arbitrary and I suggest a much more powerful (and simpler) approach would be to treat diagnostic delay as a continuous outcome in your regression models.- "Variables which are clinically or socially relevant or reach the significant level at univariate analysis will be included in the multivariate analysis" - please pre-define what variables will be considered clinically or socially relevant (this should inform the data you choose to collect). The statement about significance testing at univariate analysis is not helpful. If a factor is clinically or socially relevant you will include in multivariate model regardless of univariate analysis p-value. If a factor is not clinically or socially relevant , why would you be including in your analysis at all? Therefore, it is simplest to pre-define the clinically or socially relevant factors you intend to include. Also predefine how these factors will be categorised / analysed. Eg will age be treated as a categorical or continuous variable etc.
--

	-A small number of planned sub-group analyses should be pre-defined. -How will missing data be handled? 3. Recruitment of participants -The plan seems to be to recruit all new paediatric cancer patients in clinic over 2 years, nationally. This seems a very ambitious aim, and I would worry that it will be challenging to motivate people to recruit ALL patients in busy clinic setting over a prolonged period of time. Perhaps the authors have prior experience of running such a network study? If so please could the describe. -Who will collect the actual data? Is this done by the consultant or a research nurse? Will the study be on NIHR portfolio? -How will you maximise case ascertainment (ie ensure all eligible patients captured) and how will you assess your case ascertainment to demonstrate patients were not missed? 4. 5-year follow-up: for a very complex issue this is explained too briefly. What will be the time window for follow up (i.e. 4 yr 6 months to 5 years 6 months, or is it as soon as possible after 5 years etc)? How will patients be followed up (notes review, telephone, clinic etc - if contacted how will contact be made?). Definitions of outcomes collected? Why not collect continuous outcomes (time to relapse, time to death etc). Given the nature of the study topic, when does the clock for 5 year survival start (ie from symptom onset or diagnosis date)? Alternatively, you may wish to drop this section and prepare a separate follow up protocol in the future. Minor points -It may be helpful to include a list of included tumour types in the appendix -I can see the study has ethical approval, so this is a minor question only, but is the approach to consent correct? That is, for 16-17 year olds, should competence not be assumed (as it is for an adult) unless there is obvious reason to conduct a formal assessment - so there is no need for researchers to formally assess capacity for all 16-17 year olds (this is what I interpreted from the text as being planned)? Also there is a statement about assess competence for <16 year olds but it isn't clear how this is then used? Will children consent for themselves if competent (it seems to be implied consent will be taken from carer regardless of competence)? HRA ref: https://www.hra.nhs.uk/planning-and-improving-research/policies-standards-legislation/research-involving-children/ -On the case report form "Date of biopsy/surgery" - these should be two separate items as they are completely different issues. There should also be a 'not applicable' option for these. -Should the date treatment commenced not be collected? Good luck
--	--

REVIEWER	Phillips, Bob University of York, CRD
REVIEW RETURNED	04-Nov-2021

GENERAL COMMENTS	This is a very clear and comprehensive - and readable - protocol for a national study looking at how long it takes to get a diagnosis of cancer in those younger than 18y.
--

	There are just a couple of things within this that would benefit from improving. I refer to the PDF page numbers. p15 - Is it gender or sex that's being collected? (Boy/Girl vs Male/Female in binary.) p18 - The use of univariate "gatekeeping" multiVARIABLE .. many explanatory variables (unlikely multivariate - multiple outcomes being predicted) .. is considered to be a bit 'old hat' and could lead to the wrong answers. Non-selection for MV analysis, or selection based on a clinical/logic model to be evaluated, is probably better.
--	--

REVIEWER	Feltbower, Richard University of Leeds
REVIEW RETURNED	14-Dec-2021

GENERAL COMMENTS	This is an important study protocol focusing on one of the main priority areas for CYP with cancer and which I read with interest. In my opinion, there are several missed opportunities to align with national cancer registration work and adopt appropriate statistical methodology to undertake a robust epidemiological study.
---

VERSION 1 – AUTHOR RESPONSE

Thank you for reviewing this paper and for your important comments. We very much appreciate the time taken for such a detailed review of this protocol. We have revised the manuscript and our responses are outlined in the attached document.

VERSION 2 – REVIEW

REVIEWER	Feltbower, Richard University of Leeds
REVIEW RETURNED	17-Jan-2022

GENERAL COMMENTS	Happy with these changes, although I'd appreciate it if the authors could correct the use of 'univariate' and 'multivariate' analyses by replacing these terms with 'univariable' and 'multivariable'.
--